# Bioactive Factors in Human Breast Milk Attenuate Intestinal Inflammation during Early Life

**DOI:** 10.3390/nu12020581

**Published:** 2020-02-23

**Authors:** Julie D. Thai, Katherine E. Gregory

**Affiliations:** 1Division of Newborn Medicine, Boston Children’s Hospital, Boston, MA 02115, USA; 2Department of Pediatric Newborn Medicine, Department of Nursing, Brigham and Women’s Hospital, Boston, MA 02115, USA; kgregory1@bwh.harvard.edu

**Keywords:** human milk, breast milk, intestinal inflammation, bioactive, necrotizing enterocolitis

## Abstract

Human breast milk is well known as the ideal source of nutrition during early life, ensuring optimal growth during infancy and early childhood. Breast milk is also the source of many unique and dynamic bioactive components that play a key role in the development of the immune system. These bioactive components include essential microbes, human milk oligosaccharides (HMOs), immunoglobulins, lactoferrin and dietary polyunsaturated fatty acids. These factors all interact with intestinal commensal bacteria and/or immune cells, playing a critical role in establishment of the intestinal microbiome and ultimately influencing intestinal inflammation and gut health during early life. Exposure to breast milk has been associated with a decreased incidence and severity of necrotizing enterocolitis (NEC), a devastating disease characterized by overwhelming intestinal inflammation and high morbidity among preterm infants. For this reason, breast milk is considered a protective factor against NEC and aberrant intestinal inflammation common in preterm infants. In this review, we will describe the key microbial, immunological, and metabolic components of breast milk that have been shown to play a role in the mechanisms of intestinal inflammation and/or NEC prevention.

## 1. Introduction

Human breast milk is well known as the optimal source of nutrition during early life, as a result of a nutritional content that evolves with the needs of the growing infant [1,2]. Equally important to its nutritional attributes, human breast milk contains several bioactive factors that promote immune health, protecting against infectious and inflammatory disease processes throughout childhood [2,3,4,5,6,7,8,9,10,11]. In this review, we will focus on specific microbial, immunological and metabolic factors and the role they play in attenuating inflammation during early life. 

Inflammation is the result of a complex cascade of chemical signals released by immune cells. [12] It is a necessary and protective process of the innate immune system, required for physiological responses, such as initiating tissue repair and eliminating pathogenic insults [13]. However, evidence suggests that uncontrolled inflammation plays a prominent role in many common and chronic diseases, such as arthritis, inflammatory bowel disease, cardiovascular disease, Alzheimer’s, Parkinson’s disease, cancer, and metabolic syndrome [14]. Furthermore, inflammation early in life may lead to adverse neurodevelopmental outcomes, underscoring the importance of mitigating inflammation during the newborn period [15].

The intestine, which plays a critical role in the overall inflammatory response, is the largest immune organ in the body and, due to its large surface area, has the greatest exposure to the outside environment [16]. The newborn intestine is equipped with all the basic functional structures, but in order to fully mature, it undergoes rapid mucosal differentiation and development with exposure to enteral nutrition, namely human breast milk [16,17]. The newborn intestinal immune system is also notably immature, relying on maternal passive antibodies, particularly secretory immunoglobulin A (sIgA), for protection in the first weeks of life. In the first months of infancy, the intestinal immune system develops the ability to distinguish between foreign pathogens and safe nutrient proteins or commensal organisms [16,17,18].

The preterm infant’s intestine is more immature in structure and immune function when compared to full-term born infants, and is characterized by the elicitation of an exaggerated inflammatory response towards potential insults [19]. For example, the preterm intestine exhibits high expression of Toll-like receptor 4 (TLR4), an immune receptor expressed on leukocyte membranes that recognize molecular patterns in potential pathogens and, in turn, upregulate and suppress genes that orchestrate an inflammatory response [20,21]. This exaggerated inflammatory response has been implicated in the pathogenesis of necrotizing enterocolitis (NEC), a disease characterized by overwhelming intestinal inflammation and a major contributor to neonatal morbidity and mortality [19,22]. Breast milk has been shown to be protective against NEC in a dose-dependent manner, though the mechanism is unclear [4,23,24]. This protection is likely a result of the many bioactive components found in human breast milk that have been shown to regulate the immune system and attenuate inflammation, specifically within preterm infant intestinal biology [25,26,27,28,29] (Table 1). NEC is an extreme example of intestinal inflammation, underpinning the importance of bioactive factors in human milk. 

## 2. Methods

### Literature Search

The literature review was conducted using the PubMed and Google Scholar databases, as well as hand searches for primary studies investigating bioactive factors in human milk and their effects on intestinal inflammation. The literature search was conducted using key words, including combinations of human milk, intestinal inflammation, microbiome, *Bifdobacteria*, *Lactobacillus*, human milk oligosaccharides, immunoglobulins, secretory IgA, IgG, cytokines, growth factors, epidermal growth factor, heparin binding growth factor, vascular endothelial growth factor, lactoferrin, lactadherin, lysozyme, metabolic factors, fatty acids, antioxidants, and anti-proteases. 

## 3. Microbiome and Microbial Factors

### 3.1. Microbiome and Probiotics

Humans have evolved to develop a symbiotic relationship with the commensal bacteria comprising the intestinal microbiome. Despite the close and potentially health-compromising proximity between the many bacterial communities and the host’s intestinal surface, the host’s intestinal immune system has developed to contain and work together with the intestinal microbiota in order to maintain intestinal health homeostasis [30]. Mounting evidence suggests that exposure to commensal bacteria in early life is crucial to appropriate development of the immune system [13]. Disruption of this symbiotic relationship has been shown to lead to intestinal inflammation and disease [30]. 

The main functions of gut microbiota include facilitating the breakdown of food substances to liberate nutrients for the host to absorb, promoting host cell differentiation, protecting the host from pathogenic colonization and modulating the immune system [31]. Disruption of gut microbiota homeostasis causes shifts in microbiota balance or dysbiosis. Intestinal dysbiosis has been shown to be associated with long-term health consequences, such as obesity, diabetes and inflammatory bowel disease, as well as NEC in preterm infants [19,31]. Furthermore, germ free animals are unable to exhibit clinical signs of NEC as animals with conventional gut microbial colonization do, indicating the importance of microbial composition in the development of NEC [32]. The intestinal microbial composition of infants who develop NEC consists of unusual intestinal microbial species and overall decreased diversity of microbiota [19,33]. Therefore, the acquisition of appropriate intestinal microbiota is essential to intestinal health and prevention from inflammation and disease. 

Human breast milk has been shown to have its own unique microbiome and contains one of the main sources of bacteria to the intestine of a primarily breastfed infant [34]. Breast milk is estimated to provide 25% of a breastfed infant’s intestinal microbiota by 1 month of age as a result of exposing the infant to approximately 1 × 105 to 1 × 107 bacteria and over 700 species of bacteria daily [8,32,35]. Thus, the human milk microbiota influences the acquisition and establishment of the intestinal microbiome during infancy and is thought to be a major factor involved in innate immunity during early life [36].

The most commonly reported genera in human milk include *Staphylococcus, Streptococcus, Lactobacillus, Enterococcus, Bifidobacterium, Propionibacterium, *as well as the family *Enterobacteriaceae* [36,37,38]. However, the types and amounts of bacteria in human milk are likely impacted by many factors including genetics, maternal health and diet, stage of lactation and geographic location [39]. In a study performed by Cabrera-Rubio et al., mothers with higher body mass indexes (BMI) had higher levels of *Lactobacillus* in colostrum and lower numbers of *Bifidobacterium* in their breast milk at 6 months postpartum. Moreover, mothers who delivered via cesarean section had decreased amounts of *Leuconostocaceae*, a family of bacteria within the order of *Lactobacillus, *in their breast milk, compared to mothers who delivered vaginally [40]. 

*Bifidobacteria *have been shown to modulate inflammation and increase colonic luminal short chain fatty acid production (SCFA) in both mice and humans [41,42]. Interestingly, a lack of *Bifidobacterium *has been shown to be associated with NEC [43]. The role of *Bifidobacterium* in attenuating intestinal inflammation has been highlighted in multiple probiotic studies. In a study done by Underwood et al. in 2014, mice fed with formula and *Bifidobacterium longum* subspecies *Infantis* had decreased incidence of NEC, as well as decreased expression of pro-inflammatory mediators, interleukin (IL)-6, chemokine-1 (CXCL-1), tumor necrosis factor alpha (TNF-α), and IL-23, as well as inducible nitric oxide synthase, an important microbial pattern sensor that triggers an inflammatory response [44]. 

*Lactobacillus rhamnosus* SHA113, isolated from breast milk, was shown to inhibit multidrug resistant *Escherichia coli* (multi-drug resistant (MDR) *E. coli*) intestinal infection in vitro and in vivo, through blocking pro-inflammatory pathways and restoring homeostasis in the intestinal microbiota. Specifically, serum pro-inflammatory cytokines, TNF-α and IL-6, were reduced and anti-inflammatory cytokine IL-10 was increased in mice receiving *Lactobacillus rhamnosus* SHA113 following MDR *E. coli* infection when compared to those who received no treatment. Furthermore, *Lactobacillus rhamnosus *reversed the increased abundance of Proteobacteria observed in the MDR *E. coli* infected mice [45].

A study by Guo et al. showed that a combination of *Lactobacillus acidophilus* and *Bifidobacterium longum* species exposed to immature human enterocytes and immature human intestinal xenografts showed a decrease in pro-inflammatory mediators IL-8 and IL-6, as well as alteration of genes in the nuclear factor kappa beta (NF-κB) signaling pathway, a critical pathway in the maintenance of immune homeostasis and a strong pro-inflammatory signaling cascade [46,47]. Specifically, there was a decrease in positive inductors of the pathway, including Toll-like receptor 2 (TLR2) and TLR4 mRNA, and an increase in negative regulators—single immunoglobulin IL-1 related receptor (SIGIRR) and Toll-interacting protein (TOLLIP)—characteristic of a matured innate immune response [48]. Bacteria within breast milk can influence the intestinal microbiome, which has important roles in regulating inflammation. 

### 3.2. Human Milk Oligosaccharides and Glycans

Human milk oligosaccharides (HMOs) are within the group of glycans, which are potent antimicrobial factors in human milk. HMOs comprise an abundant and diverse component of breast milk, being not only the third largest solid component of milk, but also constituting more than 200 different structural types. Each HMO is also structurally distinct, consisting of a mixture of glucose, galactose, N-acetylglucosamine, fucose and/or sialic acid [49]. They range from three to 32 monosaccharides in size and are undigestible by the host [50,51,52]. HMOs and glycans vary by maternal genotype and change over the course of lactation [32]. Interestingly, the maternal milk of preterm infants has higher HMO concentrations than term milk [49].

HMOs have a prebiotic role, arriving to the distal intestine undigested and able to support the growth of mutualistic bacteria, specifically certain *Bifidobacteria* taxa and *Bacteroides* species. HMOs are the sole carbon source of these certain *Bifidobacterial *taxa. These microbes ferment the prebiotic glycan into small organic acids for sustenance [32]. Given the long and diverse structure of HMOs, microbial communities can act in concordance to metabolize HMOs. [31]. HMOs and glycans also act to inhibit infection by acidifying the gut lumen [32]. HMOs produce bacteriocins and organic acids which have been proved useful for preventing the growth of pathogens [49]. They also supply fucose and sialic acid [91]. HMOs and glucosaminoglycans function as pathogen-binding inhibitors that function as “decoy” receptors for pathogens that have an affinity for binding oligosaccharide receptors expressed on the infant’s intestinal surface [2,50,51,52,53]. This mechanism modulates expression of immune signaling genes, which have been shown to repress inflammation at the mucosal surface [75]. The antiadhesive activity and prebiotic activity secondarily reduce inflammation within the intestine [32].

Colostrum HMOs have been shown to modulate immune signaling pathways, including TLR3, TLR5, and IL-1β-dependent pathogen-associated molecular pathways (PAMP), and subsequently decrease acute phase inflammatory cytokine secretion. For example, 3’-galactosyllactose, directly inhibits polyinosine–polycytidylic acid, which, in turn, decreases levels of the potent proinflammatory cytokine, IL-8. Another glycan, diasialyllacto-N-tetraose (DSLNT) has been shown to suppress NEC-like inflammation in neonatal rats [32]. Supplementation with HMOs has also been shown to attenuate intestinal inflammation. In a study on preterm pigs, a formula diet consisting of HMOs was shown to decrease lipopolysaccharide-induced cytokine secretion relative to controls. Pigs who received HMO also showed higher levels of anti-inflammatory cytokines (IL-10, IL-12, TGF-β). Whether it is in promoting anti-inflammatory bacteria or directly preventing a pathogen-induced immune response, HMOs play a significant role in attenuating inflammation within the gut. 

## 4. Immunological Factors: Immunoglobulins and Immunological Proteins 

### 4.1. Immunoglobulins

Human milk provides the only source of sIgA for the first 4 weeks of life due to the lack of functioning plasma cells in the infant. sIgA is formed by cleavage of IgA in the mammary gland, allowing its release into breast milk and subsequent consumption by the infant [92]. sIgA comprises up to 80%–90% of the immunoglobulins present in breast milk and is at its highest concentrations in colostrum and in the breast milk of mothers who deliver early [65,93,94]. In contrast, other major immunoglobulin isotypes (IgM), whose roles include promoting inflammation, are present in modest or very low concentrations. Immunoglobulin E, IgE, is absent in human milk [95]. 

sIgA comprises the first line of antigen-specific immune defense and its actions are fundamentally local. They bind to commensal or pathogenic microorganisms, toxins, viruses and other antigenic materials, like lipopolysaccharide (LPS), preventing adherence and penetration into epithelium without triggering inflammatory reactions that could be harmful during early life. This phenomenon is known as immune exclusion. Because sIgA coinhabit the outer intestinal mucosal layer with commensal bacteria, its ability to effectively recognize and eliminate pathogens while, at the same time, maintaining a mutually beneficial relationship with commensal bacteria, is crucial. Interestingly, 74% of bacteria in the gut lumen are coated with sIgA [54]. Given this role, it is not surprising that sIgA influences the composition of the intestinal microbiome and, furthermore, promotes intestinal homeostasis by preventing inappropriate inflammatory responses to pathogenic microbes and nutritional antigens. 

In a recent study by Gopalakrishna et al., maternal milk-fed infants had higher percentages of IgA that were bound to bacteria compared to formula-fed infants. In addition, higher percentages of IgA bound to bacteria in the intestine of preterm infants was associated with lower rates of NEC. Furthermore, it was observed that lower levels of IgA-bound bacteria were inversely associated with abundance of enterobacteria among infants who developed NEC. Thus, IgA binding to bacteria presumably plays a protective role against NEC, likely by limiting inflammation induced by *Enterobacteriaceae*. Mice who were fed maternal milk that lacked IgA were indistinguishable from formula-fed controls, implying that maternal milk is only protective against NEC when containing IgA. While the protective mechanism by which IgA binds bacteria is unknown, it is hypothesized that IgA may limit the ability of the bacteria to gain access to the intestinal epithelium. IgA also has a role in modifying the expression of bacterial surface proteins and motility of bacteria [29]. 

Though comprising a small proportion of immunoglobulins in breastmilk, immunoglobulin G (IgG) plays an anti-inflammatory role by direct binding, opsonization and agglutination of pathogens [52,55]. IgG is mainly transferred via the placenta from mother to fetus, but IgG is also produced in the mammary gland and detected in a majority of colostrum samples of mothers, adding to the much-needed immunological protection to the vulnerable infant [56,57].

### 4.2. Cytokines and Growth Factors 

Many cytokines, including transformation growth factor beta (TGF-β), interleukin 1B (IL-1B), IL-6, I-10, IL-12, TNF-α, interferon gamma (IFN-γ), and granulocyte-macrophage colony-stimulating factor (GM-CSF) are present in human milk [18,96]. These cytokines are small proteins or peptides that act as intercellular messengers and elicit a particular response after binding to a receptor on a target cell. Responses include mediation and regulation of immunity, hematopoiesis and inflammation. IL-10, TGF-β, IL-1 receptor antagonist (IL-1RA), Tumor necrosis factor receptor I and II (TNFR I and II) all have been shown to have anti-inflammatory roles [18].

IL-10 is an important anti-inflammatory and immunoregulatory cytokine present in high concentrations in both the aqueous phase and in the lipid layer of human milk [18]. It specifically inhibits T-helper 1 (Th1) effector cell activity, natural killer cells and macrophages, resulting in immune homeostasis [58]. In a study of mice who were genetically unable to produce IL-10, an unexpected immune response was mounted to a normal intestinal microbiota in the gut. These mice ultimately developed an enterocolitis that was similar to ulcerative colitis and celiac disease in humans, emphasizing IL-10’s importance in suppressing inflammation [59,60]. Similar results were observed in IL-10-deficient mice who underwent a NEC-inducing regimen of formula feeding, hypoxia and hypothermia [61].

The TGF-β family comprise the most abundant cytokines of human milk. The highest levels are present in colostrum and decline substantially by 4-6 weeks of life [97]. They include 3 isoforms, with TGF-β2 being the most predominant. TGF-β has many immunomodulatory properties, including stimulating intestinal maturation and defense by switching immunoglobulin classes from IgM to IgA in B lymphocytes, immunoglobulin production in the mammary gland and gastrointestinal tract of the newborn, assistance with intestinal mucosal repair, and induction of oral tolerance [18]. It is known to regulate inflammation by decreasing pro-inflammatory cytokine expression. In a study of pediatric patients with Crohn’s disease, a feeding trial consisting of supplemental TGF-β resulted in decreased mucosal IL-1 mRNA and clinical remission in 79% of patients [60,62]. TGF-β inhibits naïve T cells from differentiation into Th1 and Th2 subtypes, which promote cell mediated immune responses by secreting pro-inflammatory cytokines and promote IgE and eosinophilic responses, respectively. TGF-β also helps stabilize forkhead box P3 (FOXP3) expression and maintains the differentiation of T-regulatory cells which inhibit immune responses and temper inflammation [63]. In preterm infants, lower TGF-β levels preceded NEC development, underlining the significance of TGF-β in immune and inflammation regulation [64]. In addition to essential immunoregulatory roles in the newborn, studies have suggested the association between human milk TGF-β and allergic diseases in infancy and childhood, including asthma, eczema, food allergy, and allergic rhinitis. However, a recent systemic review by Khaleva suggested that studies were too heterogenous to deduce a clear association and that larger prospective studies are needed [97]. 

Other cytokines serve as direct antagonists to pro-inflammatory signaling. IL-1 receptor antagonist (IL-1RA) is also present in human milk and limits inflammation by competing with the pro-inflammatory cytokine, IL-1, for receptor binding. In a study of rats with colitis, those fed human milk instead of formula had similar inflammatory responses compared to those who were fed formula [60]. TNFRI and II, though present in small quantities in human milk, directly bind and inhibit TNF-α, a proinflammatory cytokine produced by a wide range of immune cells [52,65]. Soluble TLR2 is present at high concentrations in breast milk and acts as a negative regulatory mechanism for cytokines and chemokines. By acting as a decoy receptor, sTLR2 has been shown to inhibit IL-8 and TNF production by monocytes stimulated with bacterial lipopeptide, which is an agonist of TLR2 [66,67]. 

Epidermal growth factor (EGF) is a peptide that is abundant in breast milk and is important for preserving intestinal barrier function, improving nutrient transport and increasing intestinal enzyme activity [68]. Studies have shown that rats with experimental NEC who were treated with EGF had decreased intestinal inflammation and particularly decreased levels of proinflammatory cytokine, IL-18, at the site of intestinal injury, as well as IL-18 mRNA levels. IL-18 dysregulation has been associated with inflammatory diseases of the small intestine. EGF has also been shown to have indirect anti-inflammatory effects by upregulation of IL-10 [68,69]. Heparin-binding epidermal growth factor-like growth factor (HB-EGF), a member of the EGF family of growth factors, also protects against intestinal injury in the developing intestine by binding to pathogenic bacteria [70,71]. In rats who underwent ischemic/reperfusion injury had less pro-inflammatory cytokine expression, particularly TNF- α and IL-6, in vivo [71,72]. 

Similarly, vascular endothelial growth factor is a glycoprotein present in breast milk, at higher levels in the colostrum and breast milk of mothers with preterm infants than those with term infants [73]. Its primary role is mediating formation of new blood vessels, a process called angiogenesis, but it has been suggested that VEGF may also have anti-inflammatory effects. In a study performed by Karatepe et al., rats induced with NEC and given subcutaneous VEGF had less villous atrophy and less intestinal edema, as well as lower TNF- α and IL-6 levels when compared to NEC-induced rats who were not treated with VEGF [74].

Pro-inflammatory cytokines, including TNF-α, IL-6, IL-8 and IFN-γ a, are also present in mother’s milk, but at lower levels than the immunoregulatory cytokines mentioned. These pro-inflammatory cytokines also decrease overtime. Certain pro-inflammatory cytokines, IL-1B, IL-6, TNF-α, are decreased in the breastmilk of preterm infants when compared to that of breastmilk of term infants [98]. This suggests that human milk cytokine composition helps regulate intestinal inflammation in newborns and may be tailored to the particular immune system needs in the infant. 

### 4.3. Lactoferrin, Lactadherin, and Lysozyme

Lactoferrin is a single-chain metal binding glycoprotein that is abundant in breast milk with highest concentrations in colostrum that decline throughout lactation [76]. It is partially degraded in the intestine and has bacteriostatic function in the intestinal mucosa of the newborn. It specifically binds to iron, preventing growth of various pathogens who are reliant on iron for further proliferation [75]. Lactoferrin has also been shown to inhibit microbial adhesion to host cells and has direct cytotoxic effects against bacteria, viruses and fungi, specifically by forming lactoferricin, a potent cationic peptide with bactericidal activity formed during digestion of lactoferrin [77]. In regard to inflammation, lactoferrin helps limit excessive immune responses by blocking many pro-inflammatory cytokines, including IL-1β, IL-6, TNF-α and IL-8, as well as suppressing free-radical activity [18,77].

Furthermore, lactoferrin has been shown to promote the growth of probiotic bacteria, regulating intestinal homeostasis. Lactoferrin supplementation in newborns has been promising, with multiple studies showing decreased NEC and late onset sepsis rates in newborns [99]. However, in a recent large randomized control trial of 2203 infants, lactoferrin supplementation did not decrease NEC or infection rates [77]. This suggests that the etiology of NEC is multifactorial and that human milk components in their collective teamwork, rather than the individual roles, protect against NEC. 

Lactadherin, also known as milk fat globule-epidermal growth factor 8, is a glycoprotein found in human milk that prevents inflammation by enhancing phagocytosis of apoptotic cells. It specifically stimulates a signaling cascade that blocks NF-κB via the TLR4 blockade, resulting in decreased inflammation [78,79]. It is also noted to promote healing during intestinal inflammation. It was shown that lactadherin administration restored enterocyte migration in septic mice, suggesting the potential benefit of lactadherin in the treatment of intestinal injuries [78].

Lysozyme is an enzyme that has primarily antibacterial effects. It degrades the outer wall of gram-positive bacteria by hydrolyzing beta 1,4 bonds from the residues of N-aceteylmuramic acid and N-acetylglucosamine. It also has been reported to have some antiviral activity [80]. In conjunction with lactoferrin, lysozyme can also kill gram negative bacteria in vitro. Lactoferrin binds to lipopolysaccharide in the outer bacterial membrane, removing it and allowing lysozyme to access and degrade the internal proteoglycan matrix of the membrane [18]. These proteins, through their direct effects on pathogens, help to prevent excessive inflammation at the intestinal surface. 

## 5. Metabolic Factors

### 5.1. Adipokines 

Adipokines are a group of mediators primarily released by adipocytes that regulate metabolic functions within adipose tissue, liver, brain and muscle, but are also present in breast milk. Furthermore, adipokines have been recently shown to attenuate intestinal inflammation by immunoregulatory mechanisms. For example, adiponectin is an adipokine available in large quantities in human milk and crosses the intestinal barrier. It has been shown to actively regulate insulin sensitivity, as well as suppressing mature macrophage function, thereby decreasing the inflammatory response. It is available in large quantities in human milk and crosses the intestinal barrier [52,81]. Interestingly, adiponectin-deficient mice exhibited more severe colitis, decreased intestinal epithelial proliferation, increased apoptosis and cellular stress when induced with dextran sulfate sodium (DSS). This colitis was reversed in vitro when adiponectin was present [100]. However, in vivo models have suggested conflicting results and require further study. 

Leptin, another adipokine, is also present in breast milk and has been implicated in infant metabolism and weight regulation. It also has immunoregulatory functions, including T cell stimulation. Interestingly, leptin is upregulated in the mesenteric fat of Crohn’s disease patients and influences the polarization of tissue macrophages towards an anti-inflammatory phenotype [81,82,83]. Leptin-deficient mice were also protected from DSS-induced colitis and leptin administration reversed disease susceptibility [84]. In addition to metabolic functions, adipokines have an immunomodulatory role that protects against intestinal inflammation. 

### 5.2. Antioxidants and Anti-Proteases

Antioxidants scavenge free radicals, or reactive oxygen species, that are produced during the normal metabolic activity of cells. Free radicals damage cells by lipid peroxidation and alteration of protein or nucleic acid structures. Antioxidants in breast milk include α-tocepherol, β-carotene, cysteine, ascorbic acid, catalase and glutathione peroxidase [60].

Inflammatory cells produce proteases, which allow them to enter the injured tissue area. Human milk contains anti-proteases, including alpha-1-antitrypsin, alpha-1-antichymotrypsin, and elastase inhibitor, which limit the ability for pathogens to enter the body, thereby limiting inflammation locally [60].

### 5.3. Dietary Fatty Acids

Fatty acid concentrations in breast milk vary considerably over the course of lactation and are likely affected by maternal diet intake. Certain structures of fatty acids have been known to alter the host inflammatory response, particularly those following infection [13]. There are three main types of fatty acids: saturated, monounsaturated and polyunsaturated, which differ according to the number of double bonds in the acyl chain structure. Omega-6 and omega-3 polyunsaturated fatty acids (PUFAs) are the two essential fatty acids in animal cells and comprise 12% to 26% and 0.8% to 3.6% of fatty acids in mature human milk, respectively [85]. Breast milk contains a high proportion of omega-3 PUFAs, which have been shown to decrease production of inflammatory cytokines. Specifically, omega-3 PUFAs decrease the activity of NF-κB, a transcription factor that induces a range of pro-inflammatory genes, including COX-2, intercellular adhesion molecule-1, vascular cell adhesion molecule-1, E-selectin, TNF-α, IL-1B, inducible nitric oxide synthase, and acute phase protein. Omega-3 PUFA binding to the nuclear receptor, peroxisome proliferator-activated receptor, PPAR-γ, has been shown to be involved in regulating immune and inflammatory responses by inhibiting the induction of inflammatory genes by LPS, IL-1B and IFN-γ [86,87]. Omega-3 PUFAs increase anti-inflammatory microbes, such as *Lactobacillus* and *Bifidobacterium* species [13,88]. They change membrane phospholipid composition by increasing arachidonic acid, an omega-6 PUFA, subsequently decreasing the systemic inflammatory response syndrome. Omega-3 PUFAS also inhibit migration of leukocytes to site of infection by lowering expression of intracellular adhesion molecule 1 on monocytes and decreasing chemotaxis in neutrophils and monocytes [13,89,90]. In addition, specialized pro-resolving mediators (SPMs), derived from omega-3 PUFAs, specifically resolve inflammation by stopping polymorphonuclear cell migration and protect against chronic inflammatory conditions, including colitis, neuroinflammation and arthritis [101]. 

Diets rich in omega-6 PUFAS have been shown to be associated with *Enterobacteriaceae* blooms, which have in turn been associated with intestinal inflammatory responses, oxidative stress and intestinal barrier dysfunction. Arachidonic acid, the most well-known omega-6 PUFA, is the origin for inflammatory mediators, such as prostaglandins, leukotrienes and thromboxanes. Similarly, diets of saturated fat have been shown to increase activation of Toll-like receptors which have been linked to increased inflammatory response and intestinal injury [13]. Supplementation of a combination of omega-3 and omega-6 PUFAs decreased the incidence of NEC and intestinal inflammation via decreased platelet activating factor (PAF)-induced TLR4 activation in mice [102,103]. 

## 6. Conclusions

Although intestinal inflammation has a vital role in the neonatal immune response, excessive inflammation may lead to decreased gastrointestinal function and injury. The most prime example is NEC, which continues to be the most devastating gastrointestinal illness and a major cause of morbidity and mortality in the newborn population. Human milk is the ideal form of nutrition and its use has been associated with decreased incidences of NEC. As outlined in this review, human milk provides a variety of protective factors that each have a role in attenuating inflammation in the intestine (Figure 1). Specific commensal bacteria, such as *Bifidobacterium* and *Lactobacillus*, directly decrease inflammatory responses in the intestine by inhibiting activation of the pro-inflammatory NF-κB signaling pathway. HMOs promote the proliferation of inflammation, regulating commensal bacteria, and act as decoy receptors for otherwise threatening pathogens. Immunoregulatory cytokines prevent the activation of pro-inflammatory signaling pathways. Antimicrobial factors, such as lactoferrin, lactadherin and lysozyme, eliminate pathogens directly. Metabolic factors, like omega-3 PUFAs, have a multifunctional role, decreasing pro-inflammatory cytokine production and signaling, while promoting the proliferation of anti-inflammatory commensals. 

Though each factor has a specific role in decreasing intestinal inflammation, the interaction among the intestinal microbiota, human milk oligosaccharides, immunological factors and metabolic components together foster optimal intestinal biology and a healthy functioning gastrointestinal tract, free of overt inflammation and infection. For breast milk-fed infants, additional inflammatory regulation is not only protective in the preterm period, but also likely has implications in decreasing risk of acquiring long-term chronic inflammatory illnesses. Though the connection between human milk’s attenuation of intestinal inflammation and the development of these chronic illnesses in adulthood is not yet clear, emerging evidence suggests that providing human milk is crucial to optimizing both short-term and long-term health outcomes for newborns. 

## Figures and Tables

**Figure 1 nutrients-12-00581-f001:**
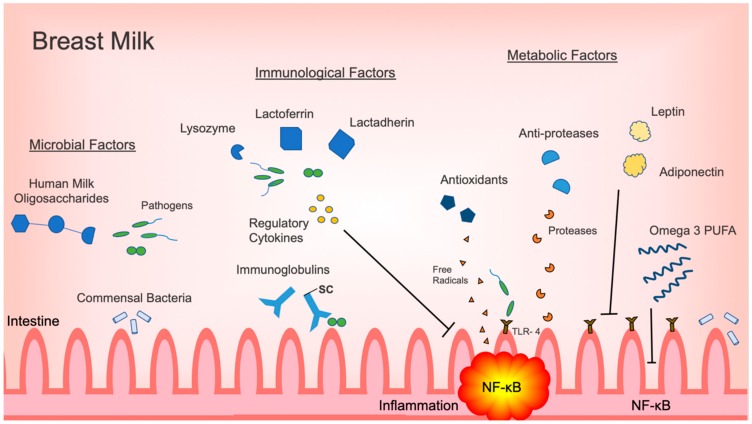
Summary of microbiologic, immunological and metabolic factors in breast milk with effects on regulating intestinal inflammation. Abbreviations: secretory component (SC); Toll-like receptor 4 (TLR4); nuclear factor kappa B (NF-κB); polyunsaturated fatty acid (PUFA).

**Table 1 nutrients-12-00581-t001:** Bioactive Components in Breast Milk and Roles in Attenuating Intestinal Inflammation.

Bioactive Components in Breast Milk	Role in Intestinal Inflammation Regulation or Prevention	Effect	References
**Microbial or microbial modulating factors**			
*Lactobacillus spp, *	-Inhibit NF-κB pathway -decrease pro-inflammatory cytokines, TNF-α, IL-6 -reverse intestinal dysbiosis in bacterial intestinal infection	-decrease inflammatory response -Restore intestinal microbiome homeostasis	[41,42,43,44]
*Bifidobacterium spp*	-increase SCFA production -Decrease pro-inflammatory CK release (IL-6, CXCL-1, TNF-α, IL-23) and iNOS	-promote anti-inflammatory commensal bacteria proliferation -decrease inflammatory response	[45,46,47,48]
Human Milk Oligosaccharides	-regulate commensal bacteria -act as decoy receptors for pathogens -modulate immune signaling pathways, TLR3, TLR5, PAMP	-promote healthy intestinal microbiota with anti-inflammatory properties -prevent and decrease inflammatory response	[32,49,50,51,52,53]
**Immunological factors**			
*Secretory IgA*	-bind to pathogens and commensal bacteria	-prevention of typical inflammatory response, or immune exclusion -influence intestinal microbiome	[29,54]
*IgG*	-opsonization, agglutination of bacteria	-prevention of typical acute inflammatory response	[52,55,56,57]
*IL-10*	-inhibit Th1, NK cell, macrophages	-provide immunoregulation and prevent inflammation	[18,58,59,60,61]
*TGF-* * β*	-inhibit differentiation of naïve T cells into Th1, Th2 cells -Stabilize FOXP3 expression	-decrease pro-inflammatory cytokine expression and inflammation -inhibit immune response and decrease inflammation	[18,60,62,63,64]
*ILRA-1* *TNFR I and II* *soluble TLR2*	-compete with IL-1 receptor for IL-1 -directly bind, inhibit TNF- α -decoy receptor to inhibit IL-8, TNF	-prevent pro-inflammatory cytokine expression and inflammation	[52,60,65,66,67]
*EGF* *HB-EGF* *VEGF*	-upregulate IL-10 expression -bind to bacteria -stimulate angiogenesis-	-decrease pro-inflammatory cytokine expression -prevent intestinal edema	[68,69,70,71,72,73,74]
Lactoferrin	-direct cytotoxicity on pathogens by forming lactoferricin -inhibit IL-1, IL-6, TNF-α, IL-8 -promote growth of probiotics	-eliminate trigger for acute inflammatory response -decrease pro-inflammatory cytokine expression and inflammation -regulate intestinal microbiome	[18,75,76,77]
Lactadherin	-enhance phagocytosis of apoptotic cells -blocks NF-κB pathway via TLR4 inhibition -promote healing during intestinal inflammation	-eliminate trigger for acute inflammatory response -prevent pro-inflammatory signaling and decreasing inflammatory response -limit degree of intestinal inflammation	[78,79]
Lysozyme	-degrades GP bacteria outer wall -kill GN bacteria with lactoferrin	-eliminate trigger for acute inflammatory response	[18,80]
**Metabolic factors**			
Adiponectin	-suppress mature macrophage function	-decrease inflammatory response	[52,81]
Leptin	-stimulates T cells -influence polarization of macrophages to anti-inflammatory phenotype	-regulate immune response and prevent inflammation	[81,82,83,84]
*Omega 3 PUFA*	-decrease NF- κB, bind to PPAR-γ -increase proliferation of *Lactobacillus* and *Bifidobacterium* -change membrane PL concentration -inhibit leukocyte migration	-downregulate pro-inflammatory genes --promote anti-inflammatory commensal bacteria proliferation -decrease degree of inflammatory response	[13,85,86,87,88,89,90]
Antioxidants	-scavenge free radicals	-prevent injury and inflammation	[60]
Anti-proteases	-metabolize proteases produced by inflammatory cells	-prevent excessive inflammatory response	[60]

Abbreviations: Nuclear factor kappa B (NF-κB); tumor necrosis factor alpha (TNF-α); interleukin (IL); short chain fatty acid (SCFA); cytokine (CK); chemokine-1 (CXCL-1); inducible nitric oxide synthase (iNOS); Toll-like receptor (TLR); pathogen-associated molecular pattern (PAMP); Immunoglobulin (Ig); T-helper (Th) cell; natural killer cell (NK); transformation growth factor beta (TGF-β); forkhead box P3 (FOXP3); interleukin receptor antagonist 1 (ILRA-1); tumor necrosis factor receptor (TNFR); epidermal growth factor (EGF); heparin-binding epidermal growth factor (HB-EGF)-like growth factor; vascular endothelial growth factor (VEGF); gram positive (GP); gram negative (GN); polyunsaturated fatty acid (PUFA); peroxisome proliferator-activated receptor gamma (PPAR-γ); phospholipid (PL).

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
