# Peer review of "Bioactive Factors in Human Breast Milk Attenuate Intestinal Inflammation during Early Life"

_nutrients, 2020, doi:10.3390/nu12020581_

Round 1
Reviewer 1 Report
Overall a very nice review of the immunomodulating factors of breast milk and how they may impact the protection of the immature intestine. Well written manuscript. I only have a few minor suggestions.
1) The manuscript is very text-heavy. A summary figure could potentially be a nice way to break up 10 pages of text. It isn't going to make or break the article but it may help as a good reference for the reader.
2) I liked the inclusion of the metabolic factors and think this is sometimes overlooked in the field. The authors may want to also consider recent works by Roghair et al (PMID 30845123 and 32000260) as they are relevant to premature infants and leptin.
3) I would also suggest for completeness sake that the authors have an "other" section which can briefly include some of the other important hormones in breast milk that may impact the environment. I'm thinking along the lines of HB-EGF, EGF, VEGF, etc... While not directly immunomodulating, they may play a role. For example, EGF can impact the epithelial proliferation which will impact Paneth and goblet cell biology and indirectly impact the biome. Plus Dvorak had a nice paper on EGF and prevention of NEC in part by MUC2 regulation (PMID 16798726). I think this could be a brief, but important addition to an otherwise very nice review.
Author Response
Dear Reviewer,
Thank you so much for your suggestions. Please see below for our responses to your comments.
"1) The manuscript is very text-heavy. A summary figure could potentially be a nice way to break up 10 pages of text. It isn't going to make or break the article but it may help as a good reference for the reader."
I do agree, this is quite long. We have edited down the introduction to about a half page to address this. I also did provide a summary figure. Please see page 8 of the revised manuscript.
"2) I liked the inclusion of the metabolic factors and think this is sometimes overlooked in the field. The authors may want to also consider recent works by Roghair et al (PMID 30845123 and 32000260) as they are relevant to premature infants and leptin."
Thank you so much for suggesting these relevant works. I did read them and do think leptin is important. I left the leptin section as it originally was as the manuscript is quite long already and I wanted to simply point out the direct effects on immunity and intestinal inflammation.
"3) I would also suggest for completeness sake that the authors have an "other" section which can briefly include some of the other important hormones in breast milk that may impact the environment. I'm thinking along the lines of HB-EGF, EGF, VEGF, etc... While not directly immunomodulating, they may play a role. For example, EGF can impact the epithelial proliferation which will impact Paneth and goblet cell biology and indirectly impact the biome. Plus Dvorak had a nice paper on EGF and prevention of NEC in part by MUC2 regulation (PMID 16798726). I think this could be a brief, but important addition to an otherwise very nice review."
I think this was a wonderful suggestion. I did originally want to include EGF as it is present in such high quantities in breast milk and in my original reading didn't come across direct effects of EGF on intestinal inflammation. But your suggestion of Dvorak allowed to find such relevant works and I have included this in the manuscript. See page 7 of the revised manuscript.
Thank you again for your suggestions. I think it really does add to this revised manuscript. Please let me know what you think.
Julie Thai

Reviewer 2 Report
In this manuscript the authors provide a well constructed descriptive review of the literature detailing the various microbial, immunological and metabolic factors within breast milk and their influence on intestinal inflammation. They emphasise the role of these factors on necrotising enterocolitis. The aim of the review is clearly and succinctly described within the abstract.
The introduction is detailed and could be shortened to provide a punchy opening to the review highlighting the key components to be covered. I would also suggest a brief statement on the methodology of the review including the databases searched and the terms of the search and how the authors selected the papers included within the review.
The authors have comprehensively detailed the key areas covered including the microbial, immunological and metabolic factors of breast milk as it related to their role in intestinal inflammation of pathological states such as necrotising enterocolitis.
The table included summarises the topics and key areas covered within the review. I would suggest also including a figure that provides a pictorial representation of some of these points.
Minor suggestions for revision
-Shorten introduction
-Brief statement on methodology for review
-Include figure if possible
-Avoid drawing conclusions such as breast milk prevents necrotising enterocolitis as this can not be concluded from the review. Can only assert that breast milk ‘may’ prevent necrotising enterocolitis
In conclusion the authors have presented a well constructed review of the literature on this topic.
Author Response
Dear Reviewer,
Thank you so much for your helpful suggestions. Please see below for our responses to your comments.
"The introduction is detailed and could be shortened to provide a punchy opening to the review highlighting the key components to be covered."
We have shortened the introduction to about half the length of its original content. We agree that it was quite long.
"I would also suggest a brief statement on the methodology of the review including the databases searched and the terms of the search and how the authors selected the papers included within the review."
We did include a very brief statement of methodology. As this is a review paper, we did not think it necessary to go too in detail.
"The table included summarises the topics and key areas covered within the review. I would suggest also including a figure that provides a pictorial representation of some of these points."
We have included a pictoral summary of the key components of the review. Please see page 8 in revised manuscript.
"Avoid drawing conclusions such as breast milk prevents necrotising enterocolitis as this can not be concluded from the review. Can only assert that breast milk ‘may’ prevent necrotising enterocolitis"
We have made some changes to the wording in the abstract and conclusion to address this.
In addition to the comments above, we did also add a section on EGF, VEGF, and HB-EGF on page 7 of the revised manuscript for completeness of the review.
Thank you again for your suggestions. Please let me know what you think.
Sincerely,
Julie Thai
